# Mosquitoes, Lymphatic Filariasis, and Public Health: A Systematic Review of *Anopheles* and *Aedes* Surveillance Strategies

**DOI:** 10.3390/pathogens12121406

**Published:** 2023-11-29

**Authors:** Arumugam Bhuvaneswari, Ananganallur Nagarajan Shriram, Kishan Hari K. Raju, Ashwani Kumar

**Affiliations:** 1Indian Council of Medical Research—Vector Control Research Centre, Puducherry 605006, India; bhuvi6face95@gmail.com (A.B.); khkraju@gmail.com (K.H.K.R.); ashwani07@gmail.com (A.K.); 2Center for Global Health Research, Saveetha Medical College and Hospital, Saveetha Institute of Medical and Technical Sciences, Saveetha University, Chennai 605102, India

**Keywords:** surveillance, *W. bancrofti*, Lymphatic Filariasis, mosquito sampling methods, sub-periodic filariasis, *Aedes* spp., *Anopheles* spp.

## Abstract

Lymphatic Filariasis (LF) affects over 120 million people in 72 countries, with sub-periodic filariasis common in the Pacific. *Wuchereria bancrofti* has three physiological races, each with a unique microfilarial periodicity, and each race is isolated to a specific geographical region. Sub-periodic *W*. *bancrofti* is transmitted by various *Aedes* mosquito species, with *Aedes polynesiensis* and *Aedes samoanus* being the primary vectors in Samoa. The *Aedes scutellaris* and *Aedes kochi* groups are also important vectors in the South Pacific Islands. Anopheles species are important vectors of filariasis in rural areas of Asia and Africa. The *Anopheles gambiae* complex, *Anopheles funestus*, and the *Anopheles punctulatus* group are the most important vectors of *W*. *bancrofti*. These vectors exhibit indoor nocturnal biting behaviour and breed in a variety of habitats, including freshwater, saltwater, and temporary water bodies. Effective vector surveillance is central to LF control and elimination programs. However, the traditional Human Landing Collection (HLC) method, while valuable, poses ethical concerns and risks to collectors. Therefore, this review critically analyses alternative trapping tools for *Aedes* and *Anopheles* vectors in LF-endemic regions. We looked at 14 research publications that discussed *W. bancrofti* vector trapping methods. Pyrethrum Spray Catches (PSC), one of the seven traps studied for *Anopheles* LF vectors, was revealed to be the second most effective strategy after HLC, successfully catching *Anopheles* vectors in Nigeria, Ghana, Togo, and Burkina Faso. The PSC method has several drawbacks, such as the likelihood of overlooking exophilic mosquitoes or underestimating *Anopheles* populations. However, exit traps offered hope for capturing exophilic mosquitoes. *Anopheles* populations could also be sampled using the Anopheles Gravid Trap (AGT). In contrast, the effectiveness of the Double Net Traps (DNT) and the CDC Light Trap (CDC LT) varied. Gravid mosquito traps like the OviArt Gravid Trap (AGT) were shown to be useful tools for identifying endophilic and exophilic vectors during the exploration of novel collection techniques. The Stealth trap (ST) was suggested for sampling *Anopheles* mosquitoes, although specimen damage may make it difficult to identify the species. Although it needs more confirmation, the Ifakara Tent Trap C design (ITT-C) showed potential for outdoor mosquito sampling in Tanzania. Furvela tent traps successfully captured a variety of *Anopheles* species and are appropriate for use in a variety of eco-epidemiological settings. By contrast, for *Aedes* LF vectors, no specific sampling tool was identified for *Aedes niveus*, necessitating further research and development. However, traps like the Duplex cone trap, Resting Bucket Trap (RB), and Sticky Resting Bucket trap (SRB) proved effective for sampling *Aedes albopictus*, offering potential alternatives to HLC. This review emphasises the value of looking into alternative trapping methods for *Aedes* and *Anopheles* vectors in the LF-endemic region. Further research is required to determine the efficacy of novel collection techniques in various contexts, even if PSC and AGT show promise for sampling Anopheles vectors. The identified traps, along with ongoing research, provide valuable contributions to vector surveillance efforts in LF-endemic regions, enabling LF control and elimination strategies to advance.

## 1. Introduction

The elimination of Lymphatic Filariasis (LF) and malaria, two mosquito-borne diseases, is planned for 2030 through preventive chemotherapy and vector control measures [1,2]. Sensitive, quick, and species-specific diagnostic methods for parasite detection in humans and vectors are required to confirm the cessation of transmission, and they are crucial for determining disease prevalence [3,4,5,6,7,8]. These technologies define intervention endpoints and confirm the efficacy of mass drug administration (MDA) programmes [9].

Medical entomologists have been utilizing molecular xenomonitoring (Mx) for decades to assess the risk of transmission of vector-borne illnesses (VBDs) by detecting human infections in arthropod vectors. Mx, a powerful tool for tracking disease transmission, as it can detect microfilarial DNA in mosquito samples even in trace amounts, served as a proxy for human infection surveillance related to VBDs [10,11,12,13,14,15,16,17]. Creating xenomonitoring systems for programmatic use that are accurate, reliable, and cost-effective can be challenging due to the diverse vector–parasite combinations. 

In 2022, the World Health Organization (WHO) estimated that there were 120 million people infected with *W. bancrofti* and 12 million people infected with *Brugia malayi*. The disease is most common in tropical and subtropical regions, and it is estimated that 80% of all cases are found in Africa (World Health Organization, Global Programme to Eliminate Lymphatic Filariasis. Progress Report 2022. Geneva: World Health Organization; 2023). Research on mosquito sampling techniques for *Anopheles* and *Aedes* vectors of diurnally sub-periodic (DspWB) and nocturnally periodic (NpWb) *W. bancrofti* (*Wb*) is lacking, despite the abundance of literature emphasizing the use of Mx for LF elimination with a *Culex quinquefasciatus* (*C.q.*)—*W. bancrofti* (*Wb*) vector–parasite combination. Choosing the best Mx tool for public health programmes and surveillance is critical, even though *Anopheles* and *Aedes* vectors are less responsible for the LF burden than *Culex*-transmitted filariasis.

The epidemiological significance of parasite DNA prevalence in local vector mosquitoes can be demonstrated by improving mosquito sampling techniques for local mosquito vector species. This will lead to sufficient sample sizes and more accurate prevalence estimates. Consequently, these enhancements have increased the operational value of LF and malaria elimination programmes.

The utilisation of different mosquito sampling techniques by vector control programmes in diverse eco-epidemiological settings may lead to biased assessments of species diversity and abundance. This systematic review critically examines the prevalence, geographic distribution, and bio-ecology of sub-periodic filariasis, specifically focusing on *Anopheles* spp. and *Aedes* spp. The review also investigates and compares the effectiveness of various Mx traps. The primary objective is to identify the most suitable Mx tool for various tasks, considering the available information on sampling techniques while taking into account the biology of the mosquito species acting as vectors.

### 1.1. Prevalence and Distribution of Lymphatic Filariasis

Over 120 million people in 72 countries across Asia, Africa, the Western Pacific, and parts of the Caribbean and South America are affected by LF. Sub-periodic filariasis caused by *W. bancrofti* is particularly prevalent in the Pacific region, including the islands of Tahiti, Samoa [18], Tonga, and Fiji, Australia, New Guinea, and the nearby Melanesia, Micronesia, and Polynesia islands are all part of the Pacific region [19].

The South Pacific islands exhibit a similar pattern of patchy filariasis distribution as the rest of the world [20,21]. Early data from Fiji [22] revealed a low frequency of 6.4% among residents of the Labasa, compared to a high prevalence of 25.2% in Taveuni. The prevalence rates in various communities appeared to be influenced by the behaviours of the population and the proximity of densely populated vector areas to human settlements, increasing the likelihood of infection transmission. Particularly in the Nancowry group of islands, Nicobar district, the day-biting *Aedes* (*Downsiomyia*) *niveus* transmit the diurnally sub-periodic *W. bancrofti* disease in India [23,24,25,26,27,28].

#### 1.1.1. Physiological Races of *W. bancrofti*

There are three physiological races of *W. bancrofti*, each having a unique microfilarial periodicity. Apart from the Polynesian sub-region, the nocturnally periodic race is widely distributed in tropical and sub-tropical environments worldwide. A nocturnally sub-periodic race is found in the jungle regions of West Thailand, while the diurnally sub-periodic race is isolated to the Polynesian sub-region [29,30,31]. Each race has unique intermediate hosts, and the microfilarial periodicity of each race coincides with the biting rhythm of the principal vector mosquitoes.

#### 1.1.2. Types of *W. bancrofti* Infection Identified, Based on Their Ecological Distribution 

The *Culex fatigans* type, transmitted by the *Culex pipiens* complex, including races like *Culex fatigans*, is known as *Culex quinquefasciatus and Culex molestus*. This is the most widely distributed ecological type [29].The *Anopheles* type, in tropical Africa, *Anopheles gambiae*, *Anopheles funestus*, and related species are the principal vectors of *W. bancrofti* in rural areas, while other regions have vectors such as *Anopheles maculatus*, *Anopheles whartoni*, *Anopheles flavirostris*, *and Anopheles punctulatus* [29].The *Aedes* (*Finlaya*) *poecilus* type is responsible for transmitting nocturnally periodic *W*. *bancrofti* in the Phillipines. Additionally, the *Aedes (Finlaya)* kochi group serves as efficient vectors for the diurnally sub-periodic race in the Polynesian region.The *Aedes* (*Ochlerotatus*) *vigilax* type is the primary vector of the diurnally sub-periodic race of *W*. *bancrofti* endemic in the New Caledonian region.*Aedes* (*Stegomyia*) *polynesiensis*-type mosquitoes are the principal vectors of the diurnally sub-periodic form *W*. *bancrofti* in the Polynesian region.

##### Breeding Ecology, Biology and Implication in the Transmission 

In Samoa, sub-periodic *W*. *bancrofti* is primarily transmitted by two vectors: *A. polynesiensis* and *A. samoanus* [12,32,33,34]. *A*. *polynesiensis*, a container breeder, and *A*. *samoanus*, a leaf axil breeder, are nocturnal species restricted to the Samoa islands [18,35,36,37,38]. Another member of the kochi group, *A*. *tutuilae*, breeds exclusively in pandanus leaf axils and is also a nocturnal species found only in Samoa [18]. 

In the South Pacific Islands, the sub-periodic *W. bancrofti* is primarily transmitted by *A*. *pseudoscutellaris*, and along with *A. polynesiensis*, they were efficient transmitters, and the night-biting *A*. *fijiensis* of the kochi group was equally efficient [22,39,40]. *C*. *quinquefasciatus*, the vector of periodic *W. bancrofti*, could also transmit sub-periodic *W*. *bancrofti* to a limited extent in the Society Islands and Fiji [22,40]. Further studies by Burnett [41], Rossen [42], and Symes [22,40] confirmed these findings, emphasizing the need for re-examination potential vectors, particularly in areas where members of the *A. scutellaris* and *kochi* groups have been reported [43].

In India’s Andaman and Nicobar Islands, DspWb is transmitted by the day-biting *Aedes* (*Downsiomyia*) *niveus*, a tree-hole breeder, with diurnal biting behaviour. *A. polynesiensis* in American Samoa and Western Samoa exhibits similar biting behaviour, primarily diurnal with a small proportion biting at night [27,44]. Sampling *A. niveus* presents a significant challenge for researchers and LF control programmes conducting surveillance [45].

In Samoa, efficient vectors of sub-periodic *W. bancrofti* include *A. polynesiensis*, *A. upolensis*, and in the Tongo region *A. tabu. A. polynesiensis* prefers resting in dry coconut husks, tree holes, the undersides of partially detached bark on dead trees, and similar sheltered sites. *A. upolensis* is a true forest dweller, diurnally active and breeds in tree hollows or cavities of dead trees. *A. tabu* is predominantly found in plantations but also occurs in shady areas in villages, limited to the Tonga islands. Their breeding habitats include tree holes, artificial containers, coconut shells, and leaf axils of taro [35] Other important *Aedes* vectors are *A. poecilus*, the *A. scutellaris* group, and *O. togoi* (earlier known as *A. togoi*) [46].

##### The Anopheles Vector and Its Ecology

Anopheles species are significant vectors of malaria and filariasis in rural regions of Asia and Africa. Some examples include the *A. punctulatus* group in Papua New Guinea (PNG) [47], *A. gambiae s.s.*, *A. arabiensis*, and *A. funestus* on the Kenyan coast [48]; *A. subpictus* in the Indonesian islands of Flores and Timor [49]; and *A. gambiae s.s.* in Ghana [50], which transmits both *P. falciparum* and *W. bancrofti* parasites.

There are about 26 *Anopheles* species vectoring Bancroftian and *B. filariasis*. Of these, eighteen species transmit *W. bancrofti*, three species transmit *B. malayi*, and five species transmit both parasites. *A. barbirostris* is the only known vector of *B. timori.* Among them, the *A. funestus* group and members of the *A. gambiae* complex including *A. gambiae s.s.*, *A. arabiensis*, *A. melas*, and *A*. *merus* are the most important vectors of *W. bancrofti. A. merus* breeds in saltwater, while the other three species breed in freshwater [49] with *A. melas* often associated with mangroves [51].

In Africa, the *A. gambiae* complex and *A. funestus* are the most important vectors of *W. bancrofti* [52]. These vectors exhibit indoor nocturnal biting behaviour. *A. gambiae* is highly anthropophilic and employs a “patrolling” or “ranging” flight strategy to encounter host cues. *A. funestus* is highly endophilic and anthropophilic but can display moderate to high zoophagy in areas with large livestock populations. They breed throughout the year, with *A. funestus* preferring permanent water bodies and certain stagnant water bodies, while the *A. gambiae* complex breeds in temporary or man-made water bodies like pools, puddles, brick pits, fields, construction sites, hoof prints, or tire tracks. This adaptability allows them to maintain population numbers even during dry months, promoting year-round malaria transmission [53].

In Asia, *W. bancrofti* is transmitted by *A. jeyporiensis candidiensis* and *A. minimus* in China, by *A. flavirostris* in the Philippines, and by *A. balabacensis*, *A. maculatus*, *A. letifer*, and *A. whartoni* in Malaysia [49]. The *A. punctulatus* group includes *A. punctulatus*, *A. farauti*, and *A. koliensis*, which are the principal vectors of the periodic *W. bancrofti* in Papua New Guinea (PNG), West Papua (Indonesia), Solomon Islands, and Vanuatu [54,55].

The *A. punctulatus* group prefers to breed in small, shallow, exposed pools devoid of other flora and fauna [56]. In an inland village in PNG, 99.9% of *A. punctulatus* were recorded. *A. farauti* is a coastal species that can breed in brackish water but is also found at altitudes over 1000 m above sea level in PNG. *A. koliensis* is a nocturnal mosquito with a preference for indoor feeding and breeds in streams at the forest margins. In East Sepik Province, PNG, both *A. punctulatus* and *A. koliensis* were found to be potential vectors. *A. punctulatus* breeds along river edges during the rainy season and, during the dry season, along sections of the dried-up river, forming numerous sun-lit puddles that serve as additional breeding sites [57]. *Aedes* and *Anopheles* species which are responsible for transmitting *W. bancrofti* in various regions of the world is listed below (Table 1)

## 2. Methodology

### 2.1. Database Search and Systematic Review

We systematically analysed published research articles using specific databases including Google Scholar, ResearchGate, PubMed, and ScienceDirect. Our search focused on topics such as molecular xenomonitoring, *W. bancrofti* (*Wb*), Lymphatic Filariasis, Mosquito sampling methods, mosquito trapping techniques, *Aedes-* and *Anopheles*-transmitted filariasis, and sub-periodic filariasis. Through this search, we identified 41 relevant articles. 

In our analysis, we specifically examined the efficiency of different mosquito trapping techniques, with a focus on sampling *Aedes* and *Anopheles* mosquito vectors of *W. bancrofti*. This information is crucial for conducting molecular xenomonitoring, VBD surveillance, and implementing effective public health programmes. After careful evaluation, we selected 14 records that met our inclusion criteria, making them eligible for data collection and comprehensive analysis.

### 2.2. Exclusion Criteria

Brugian filariasis studies involved the *Anopheles* species in transmission. LF was transmitted by *Culex* mosquitoes.

Studies focused on human blood surveys for microfilarial infection detection for Transmission Assessment Surveys (TAS). 

### 2.3. Inclusion Criteria

*Aedes* and *Anopheles* transmitted *W. bancrofti*.Co-endemicity of LF and malaria transmitted by the *Anopheles* vector.Molecular xenomonitoring of *Aedes* and *Anopheles* LF vectors.Mosquito trapping techniques employed in diverse LF-endemic regions for Transmission Assessment SurveysEfficiency assessment of various mosquito traps.

The flow chart below depicts the search strategies for sampling techniques related to *Anopheles-* and *Aedes*-mediated *W. bancrofti*. Full-text records meeting the inclusion criteria were selected for review. The study design was described using all five steps of the PRISMA (Preferred Reporting Items for Systematic Reviews) (Figure 1) checklist to ensure review quality.

## 3. Sampling Strategies for LF-Endemic *Aedes* and *Anopheles* Vectors

The 14 included publications focused on assessing trapping techniques for *Aedes* and *Anopheles* vectors transmitting *W. bancrofti* in endemic regions where an Mx of LF has been conducted. While the HLC method has been essential, ethical concerns and risks to collectors necessitate exploring alternative trapping tools [51]. 

This review critically analysed the efficiency of seven traps for sampling Anopheles LF vectors such as PSC, GT, BGS, CDC LT, ET, DNT, and AGT (Table 2) and discussed novel collection methods including the Stealth trap, Ifakara Tent Trap, Mbita trap, Furvela tent trap as alternatives to HLC [51]. For Aedes LF vectors, four different traps were analysed (BGS, GT, CDC LT, DNT) along with the Resting Bucket Trap (RB), Sticky Resting Bucket trap (SRB), Duplex cone trap, and novel sticky trap [51].

PSC was the second most efficient method after HLC, effectively capturing *Anopheles* vectors in Nigeria, Ghana, Togo, and Burkina Faso [7,51,69,79]. However, PSC has drawbacks, as it may miss exophilic mosquitoes or underestimate *Anopheles* populations [72,78,80,81]. The Exit trap was useful for trapping exophilic mosquitoes [71]. BGS showed varying results in different settings, being more effective in some regions but less so in others [17,34,50]. The AGT appeared to be an appropriate trap for sampling *Anopheles* populations [51]. CDC LT and DNT also had variable efficiency [71]. In Mali, HLC was observed to be more productive than the PSC [82].

The review identifies traps with potential for xenomonitoring LF vectors, but further studies are needed to assess their effectiveness in different settings [51,73]. The CDC Light Trap captured more *Culex* and fewer *Anopheles* vectors [51]. A comparison of various vector sampling techniques in *Anopheles-* and *Aedes*-mediated LF-endemic regions is provided in Table 3.

## 4. Methods for Sampling Mosquitoes to Conduct Disease Surveillance and Research

It is critical to comprehend the behaviour, distribution, and function of mosquitoes as disease vectors in order to effectively combat illnesses such as LF, malaria, dengue, Zika, and others. Scientists are able to advance disease management by gaining insights into mosquito populations and their interactions with the environment through the use of a variety of collection techniques.

**Human Landing Catch (HLC):** Employing mosquitoes’ biological propensity to seek a blood meal for reproduction, this technique captures mosquitoes as they descend onto human or animal hosts. They are captured using an oral aspirator prior to biting the host [83].

**Pyrethrum Spray Collection (PSC):** PSC entails the application of an insecticide aerosol containing pyrethrum within confined spaces. Mosquitoes are rendered immobile and disoriented, causing them to descend onto a white cloth, from which they can be collected [84].

**CDC Light Trap:** These devices attract and capture adult mosquitoes using artificial light sources. By simulating natural light sources, the trap attracts mosquitoes for further study [85].

**CDC Gravid Trap**: Specialized equipment built to capture female mosquitoes in search of sites where they can lay their eggs. These devices leverage the inherent behaviour of mosquitoes by employing attractants such as organic matter to establish an optimal environment for oviposition [86].

**Biogents Sentinel (BGS) Trap:** These traps imitate human or animal hosts by combining chemical attractants, visual signals, and heat. The chemical lures and design of the trap increase its attraction to blood-seeking mosquitoes [86].

**Window Exit Trap:** As mosquitoes enter or leave designated areas, this trapping system captures them. It is comprised of two chambers, one of which is baited with heat and carbon dioxide to simulate human presence and entice mosquitoes to enter; these mosquitoes are then collected in the exit chamber [87].

**DNT:** This consists of two box nets, one protecting the collector and a second larger net, which is placed directly over the inner net. The outer net is raised off the ground so that mosquitoes attracted to the human bait are collected between the two nets [88].

### 4.1. Novel Tools for Sampling Anopheles Vectors: W. bancrofti and P. falciparum Transmission

Gravid mosquito traps, such as the OviArt Gravid Trap (AGT), offer valuable sampling tools for both endophilic and exophilic vectors, enhancing the detection of parasite-infected mosquitoes for VBD surveillance and Transmission Assessment Surveys. AGT, specifically targeting gravid *Anopheles* mosquitoes, has shown promising results with improved catch size compared to other traps like the Box gravid trap. However, further improvements in battery protection and transportation convenience are needed. AGT is made of a rectangular basin measuring 45 cm × 33 cm × 11.5 cm (length × width × height), with a 4 cm hole on the side and a 6 L rectangular basin. An open plastic tube (collection chamber) was inserted into the hole and the other opening of the tube was sealed with fibreglass netting to prevent trapped mosquitoes from escaping. The tube was placed and secured halfway into an aluminium collapsible pipe. The flexible tube was connected to a 12 V fan that provided suction on the water surface [89].

The Stealth trap (ST) has been recommended as a valuable tool for capturing *Anopheles* mosquitoes, particularly *A. gambiae* s.l., in West Africa. While ST shows high capture rates, it may cause damage to specimens, making species identification challenging [82]. The Ifakara Tent Trap C design (ITT-C) has been considered promising for outdoor mosquito sampling in Tanzania and has been evaluated for routine malaria vector surveillance. Modifications and validation in different settings are needed for optimal performance [90]. The ITT has been reported to be comparatively superior in terms of capture rates compared to HLC [91] and CDC LT [92]. However, operators’ exposure to mosquito bites necessitates modifications for improved performance [93].

Furvela tent traps provide an efficient way to capture *Anopheles* mosquitoes, especially in situations with diverse mosquito fauna. Combining CDC-LT or window-exit traps (to sample endophagic mosquitoes) with Furvela tent traps (to sample exophagic ones) allows robust sampling of diverse mosquito species [94]. Both CDC-LT and Furvela tent traps are portable and suitable for surveillance. These tools hold promise for effective vector monitoring in various eco-epidemiological settings [82,90,94,95].

### 4.2. Aedes Mosquito Sampling Techniques: W. bancrofti Transmission

In a typical endemic setting for DspWb mediated by *A. niveus*, BGS, DNT, GT, and HLC were deployed for vector mosquito sampling to assess vector infection. However, none of these trapping methods were suitable for sampling *A*. *niveus*, with BGS and DNT capturing more *A. albopictus* and *A. aegypti*, and GT capturing *C. quinquefasciatus* [61]. Thus, there is currently no specific sampling tool identified for *A. niveus* apart from HLC.

In contrast, BGS was found to sample adequate numbers of *A. polynesiensis* and *A. samoanus* vectors of LF in Samoa. However, the sampling method suitable for *A. polynesiensis* in Samoa may not be applicable for *A. niveus* in Nancowry Islands, and BGS showed limited efficiency in capturing *A. niveus* [16]. For *A. albopictus*, some traps like the Duplex cone trap, Resting Bucket Trap (RB), and Sticky Resting Bucket (SRB) have been reported to be efficient in sampling.

The Duplex cone trap was found to be the most productive trap for sampling *A. albopictus*, offering a promising alternative to HLC [96]. RB and SRB traps were also effective in capturing *A. albopictus* in various habitats, with SRB showing higher capture rates [97]. Additionally, the novel sticky trap was reported to be more precise than the ovitrap for sampling *A. albopictus* in urban settings [98].

These findings suggest that, for specific mosquito species like *A. niveus*, further research and development of the suitable sampling tools are needed. Meanwhile, traps like the Duplex cone trap, RB, SRB, and novel sticky trap show promise for efficient sampling of *A. albopictus.*

## 5. Conclusions

In conclusion, this review has focused on the assessment of trapping techniques for *Aedes* and *Anopheles* vectors responsible for transmitting *W. bancrofti* in LF regions endemic to LF. While the Human Landing Collection (HLC) method has been instrumental, it raises ethical concerns and poses risks to collectors, underscoring the need for alternative trapping methods. Among the traps examined, the Pyrethrum Spray Catches (PSC) method demonstrated high efficiency in capturing *Anopheles* vectors across multiple countries. However, it may not effectively capture exophilic mosquitoes and could underestimate *Anopheles* populations. Other traps like the Anopheles Gravid Trap (AGT) and the Exit Trap, also showed potential, although their effectiveness varied in different settings. 

In the case of *Aedes* vectors, traps like BGS and CDC LT proved useful in specific regions, yet no dedicated sampling tool was identified for *A. niveus* apart from HLC. Novel traps like the Stealth trap, the Ifakara Tent Trap C design (ITT-C), and Furvela tent traps offer promising alternatives for capturing Anopheles mosquitoes in diverse eco-epidemiological settings. Regarding *A. albopictus*, traps like the Duplex cone trap, Resting Bucket Trap (RB), and Sticky Resting Bucket (SRB) exhibited high efficiency in various habitats. Nonetheless, further research is imperative to develop suitable sampling tools for specific mosquito species like *A. niveus*. In summary, these findings provide valuable insights into efficient sampling strategies for LF-endemic *Aedes* and *Anopheles* vectors, thereby facilitating vector surveillance and enhancing disease control efforts.

## Figures and Tables

**Figure 1 pathogens-12-01406-f001:**
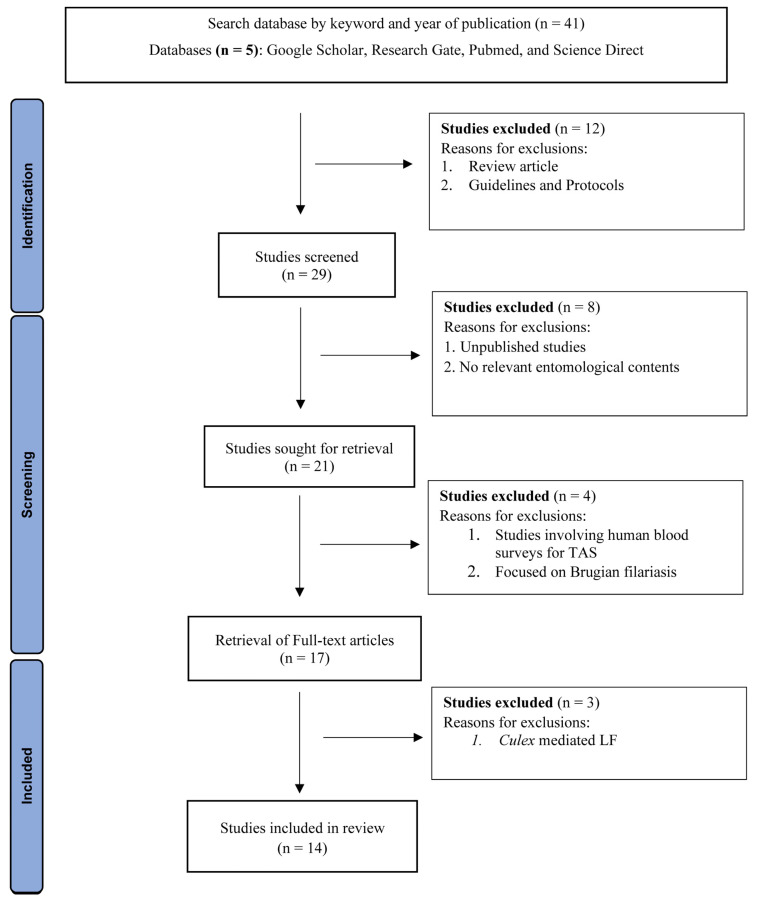
Identification of Sampling techniques for *Anopheles-* and *Aedes*-mediated *W. bancrofti* through databases. PRISMA flow diagram of study selection and inclusion.

**Table 1 pathogens-12-01406-t001:** *Aedes* and *Anopheles* LF vectors.

SL.NO.	Region	Vector Species	Reference
1	Flores and Timor (Indonesian Islands)	*An. subpictus*	WHO-2022 [3]
2	China	*An. jeyporiensis candidiensis*	
3		*An. minimus*	
4	Philippines	*An. flavirostris*	WHO-2022 [3]
5	Ghana	*An. gambiae* complex,	Owusu et al., 2015 [7]
6		*An. funestus*,	
7		*An. arabiensis*	
8		*An. melas*	
9	American Samoa	*Ae. polynesiensis*	Schmaedick et al., 2014 [12]
10		*Ae. samoanus*	
11		*Ae. aegypti*,	
12		*Ae. (Finlaya)* group	
13	India (A &N Islands)	*Ae. (Downsiomyia) niveus*	Premkumar et al., 2020 [16]
14	Thailand	*Ae. niveus* group	Harinasuta et al., 1970 [30]
15	Samoa	*Ae. polynesiensis*	Hapairai et al., 2015 [34]
16		*Ae. samoanus*	
17		*Ae. (Finlaya)* spp.	
18		*Ae. aegypti*	
19		*Ae. upolensis*	Ramalingam et al., 1968 [35]
20	Polynesian region	*Ae. kochi* group	Burnett et al., 1960 [41]
21	Philippines	*Ae. poecilus*	Bockarie et al., 2009 [46]
22	Papua New Guinea, West Papua (Indonesia), Solomon isalnds, Vanautu	*An. punctulatus*	Webber et al., 1977, 1979, 1991 [54]
23		*An. farauti*	
24		*An. koliensis*	
25	Tanzania	*An. merus*	Bartilol et al., 2021 [58]
26	Polynesia, New Caledonia	*Ae. polynesiensis*	Strickland Hunter’s Tropical Medicine and emerging Infectious diseases [59]
27		*Ae. tabu*	
28		*Ae. vigilax*	
29	Thailand	*Ae. annandalei*	Jitpakdi et al., 1998 [60]
30		*Ae. desmotes*	Gould et al., 1982 [61]
31		*Ae. harinasutai*	
32	Malaysia	*An. leucosphyrus*	Muturi et al., 2008 [62]
33		*An. barbirostris*	
34		*An. balabacensis*	
35		*An. maculatus*	
36		*An. letifer*	
37		*An. whartoni*	
38		*An. donaldi*	
39		*An. campestris*	
40	Indonesia	*An. balabacensis*	
42	Solomons island	*An. koriensis*	
43	China and Korea	*An. sinensis*	
44	Philippines	*An. minimus*	
45	Banggi Island, Sabah, Malaysia	*An. flavirostris*	Hii et al., 1975 [63]
46	Hainan island, China	*An. minimus*	Chen et al., 2002 [64]
47	Papua New Guinea	*An. koliensis*	Manguin et al., 2010 [65]
48		*An. bancroftii*	
49		*An. farauti* s.l.	
50		*An. punctulatus*	
51	Brazil, Dominican republic, Guyana, Haiti, Costa Rica, Suriname, Trinidad, Tobago and Brazil	*An. darlingi*	
52		*An. aquasalis*	
53		*An. albimanus*	
54		*An. bellator*	
55	Borneo	*An. balabacensis*	Sallum et al., 2005 [66]
56		*An. latens*	
57	Uganda	*An. bwambae*	GB White et al., 1985 [67]
58	Sri Lanka	*An. (Cellia) jamesii*	Abeyewickreme et al., 1991 [68]
59	Nigeria	*An. gambiae* s.l.	Richards et al., 2011 [69]
60		*An. funestus*	
61	Togo	*An. gambiae*	Dorkinoo et al., 2018 [70]
62	Burkina Faso (West africa)	*An. gambiae* s.l.	Sanata Coulibaly et al., 2022 [71]
63		*An. funestus* s.l.,	
64		*An. coluzzii*	
65		*An. gambiae*	
66		*An. nili*	
67	Mali	*An. gambiae* complex,	Coulibaly et al., 2015 [72]
68		*An. funestus* complex	
69	Tanzania	*An. gambiae* complex	Jones et al., 2018 [73]
70	Kenya Coast	*An. gambiae* s.s.	Mathenge et al., 2005 [74]
71		*An. arabiensis*	
72		*An. funestus*	
73	Indonesia	*An. aconitus*	Atmosoedjono et al., 1977 [75]

**Table 2 pathogens-12-01406-t002:** Studies carried out in different LF-endemic regions where the principal vectors are *Aedes* and *Anopheles*.

S. No.	Filariasis Endemic Countries	Main Vector	Context	Study Date	Study Design	Vector Sampling Methods	Sample Size	Analysis Method	Reference
1	Nigeria	*A. gambiae* s.l., *A. funestus*, *Anopheles* spp., *Culex* spp.	Post-MDA surveillance	2009	Longitudinal	PSC	4398	Dissection	Richards et al., 2011 [69]
2	Papua New Guinea	*A. punctulatus* *A. koliensis* * *A. hinesorum* * *A. farauti 4* * *A. farauti* sensu stricto.*	Post-MDA surveillance	2007–2008	Longitudinal	HLC	21,899	PCR	Reimer et al., 2013 [76]
3	American Samoa	*A. polynesiensis* *A. samoanus* *A. aegypti* *A. (Finlaya)* group *A. oceanicus* * *A. samoanus* * *A. tutuilae* * *A. nocturnus* * *C. annulirostris* * *C. sitiens* * *C. quinquefasciatus*	Post-MDA surveillance	2011	Cross-sectional	BGS + lure	22,014	PCR	Schmaedick et al., 2014 [12]
4	Mali	*A. gambiae* complex, *A. funestus* complex	Post-MDA surveillance	2007	Longitudinal	HLC	4680	Dissection	Coulibaly et al., 2015 [72]
5	Ghana	*Anopheles* spp. *Culex* spp.	Post-MDA surveillance	2008	Cross-sectional	PSC and GT	4500	PCR	Owusu et al., 2015 [7]
6	Samoa	*A. polynesiensis* *A. (Finlaya)* sp. *A. aegypti* * *A. upolensis* * *C. annulirostris* * *C. quinquefasciatus* *	Post-MDA surveillance	2012	Cross-sectional	BGS + Lure, HBC and CDC LT	5360	PCR	Hapairai et al., 2015 [34]
7	Mali	*A. gambiae* complex *A. funestus* complex *A. pharaoensis* * *A. rufipes* *	Post-MDA surveillance	2009–2013	Longitudinal	HLC and PSC	14,539	Dissection and PCR	Coulibaly et al., 2016 [77]
8	Togo	*A. gambiae*, *Culex* spp., *A. aegypti* *, *Mansonia* spp. *	Post-MDA surveillance	2015	Cross-sectional	PSC, HLC, and ET	10,872	PCR	Dorkinoo et al., 2018 [70]
9	Tanzania	*A. gambiae* complex, *C. quinquefasciatus*	Post-MDA surveillance	2015	Cross-sectional	CDC LT and GT	1650	Dissection and PCR	Jones et al., 2018 [73]
10	Ghana	*A. gambiae* complex, *A. funestus*, *A. arabiensis* *A. melas*, *A. rufipes* * *A. coustani* * *Aedes* spp * *Culex* spp.* *Mansonia* spp.*	Post-MDA surveillance	2016–2017	Cross-sectional	AGT, Box GT, CDC GT, LT, ET, BGS, IRC, PSC	2188	PCR	Opoku et al., 2018 [51]
11	India (A and N Islands)	*A. (Downsiomyia) niveus*, *C. quinquefasciatus* *A. albopictus* * *A. aegypti* * *A. edwardsi* * *A. malayensis* * *A. subalbatus **	Post-MDA surveillance	2014–2015	Cross-sectional	BGS, GT, DNT, and HLC	2170	RT-PCR	Premkumar et al., 2020 [16]
12	Burkina Faso (West Africa)	*A. gambiae* s.l., *A. funestus* s.l., *A. coluzzii*, *A. gambiae*, *A. nili*, *A. arabiensis* *	Monitoring of LF and malaria prevalence	2014 and 2015	Cross-sectional	HLC, PSC	29,183	Conventional PCR and LAMP	Sanata Coulibaly et al., 2021 [78]
13	Samoa	*A. polynesiensis*, *A. samoanus*, *A. (finlaya)* spp. *A. aegypti*, *A. albopictus* * *A. upolensis* * *C. quinquefasciatus* *	Post-MDA Surveillance	2018 and 2019	Longitudinal	BGS + Lure	13,700	PCR	McPherson et al., 2022 [17]
14	Burkina Faso	*A. gambiae*, *A. coluzzi* * *A. arabiensis* * *A. funestus* group, *A. nili*	Assessment of VSM	2018	Cross-SectionalCross-sectional	HLC, Window ET, DNT, PSC	3322	PCR	Sanata Coulibaly et al., 2022 [71]

* Non—LF vectors collected during sampling. HLC—Human Landing Catch, PSC—Pyrethrum Spray Catch, HBC—Human Bait Catch, BGS—Biogents Sentinel trap, CDC LT—Centres for Disease Control Light trap, ET—Exit Trap, GT—Gravid Trap, AGT—Anopheles gravid trap, DNT—Human baited double bed net-traps, BOX—box gravid trap, IRC—Indoor Resting Collection.

**Table 3 pathogens-12-01406-t003:** Applicability of different vector sampling techniques to sample *Anopheles* and *Aedes* vectors in LF-endemic regions apart from the Human Landing Collection.

LF-Endemic Region	LF Vector	PSC	GT	BGS	CDC LT	ET	DNT	AGT
Nigeria	*Anopheles* sp.	**✔**	**X**	-	-	-	-	-
Ghana		**✔**	**X**	**X**	**X**	**✔**	-	**✔**
Togo		**✔**	-	-	-	**✔**	-	-
Burkina Faso		**✔**	-	-	-	**✔**	**✔**	-
Mali		**X**	-	-	-	-	-	-
Tanzania		-	**X**	-	**X**	-	-	-
Samoa	*Aedes* sp.	-	-	**✔**	**✔**	-	-	-
Nancowry Islands, India		-	**X**	**X**	-	-	**X**	-

**PSC**—Pyrethrum Spray Catch, HBC—Human Bait Catch, **GT**—Gravid Trap, **BGS**—Biogents Sentinel trap, **CDC LT**—Centres for Disease Control Light trap, **ET**—Exit Trap, **DNT**—Human baited Double Net Traps, **AGT**—Anopheles gravid trap. **✔**: effective; **X**: invalid.

## Data Availability

Data is presented in tables within the manuscript.

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
