# Peer review of "Mosquitoes, Lymphatic Filariasis, and Public Health: A Systematic Review of Anopheles and Aedes Surveillance Strategies"

_pathogens, 2023, doi:10.3390/pathogens12121406_

Round 1

Reviewer 1 Report

Comments and Suggestions for Authors

This manuscript, titled "Exploring Mosquito Sampling in Lymphatic Filariasis: A Systematic Review of Anopheles and Aedes Surveillance," holds significant importance within the context of The Global Lymphatic Filariasis Elimination Programme's history. Its utility in addressing last-mile challenges encountered in the final stages, determining the elimination endpoint, and conducting post-elimination surveillance is critical.

The study rationale is meticulously documented and well-supported. Xeno-monitoring emerges as an essential component in enhancing data quality for making decisions regarding breaking disease transmission and post-MDA surveillance. The careful selection and deployment of appropriate entomological tools are indispensable to ensuring the utmost efficiency and effectiveness in the methods and tools employed, consequently strengthening the evidence for critical surveillance decisions, as articulated in this systematic review.

The methodology employed in conducting this review is well-documented and leads to the subsequent sections, including the analysis, results, discussions, and conclusions. However, it would be valuable if the authors could highlight the rationale behind excluding articles related to Brugian filariasis studies involving Anopheles species in transmission and lymphatic filariasis transmitted by Culex mosquitoes. Moreover, the limitation of the database searches to only five databases for a systematic review raises the question of whether a broader search scope could have been considered.

The flow diagram used to identify study articles for the systematic review is simple, presenting a straightforward and easy-to-understand pathway. Given that sample sizes, as found in various study articles, represent essential data for assessing the effectiveness and efficiency of distinct methods, it would be beneficial to include an additional table that details the different techniques and the corresponding vector sample sizes achieved in various regions.

While the manuscript provides an overview of various mosquito trapping techniques, a more detailed description of these methods would offer deeper insights and a better understanding of their utility. Although the study assesses trapping techniques for Aedes and Anopheles mosquito vectors, a more thorough description of these methods, along with data on the number of mosquitoes captured using each, could be presented in a more organized manner to enhance understanding and facilitate recommendations for improvement. The Human Landing Collection method appears to be the most effective; hence, it might be pertinent to recommend improvements to mitigate the associated risks to collectors while also exploring alternative approaches.

In conclusion, this manuscript is informative and raises important research questions. However, it requires addressing the issues discussed and some minor editorial refinement. I recommend the article for publication once these minor concerns are addressed.

I would like to express my gratitude for the opportunity to review this manuscript.

Comments on the Quality of English Language

The quality of English is good but for minor editorial work that is required. 

Author Response

We express our sincere gratitude to the reviewers for their invaluable suggestions. We have thoughtfully reviewed all of their recommendations and made the necessary revisions to the manuscript accordingly.

Comment 1: However, it would be valuable if the authors could highlight the rationale behind excluding articles related to Brugian filariasis studies involving Anopheles species in transmission and lymphatic filariasis transmitted by Culex mosquitoes. Moreover, the limitation of the database searches to only five databases for a systematic review raises the question of whether a broader search scope could have been considered.

Response:

While Culex mosquitoes are the primary vectors for lymphatic filariasis (LF), Anopheles and Aedes vectors, though less significant in LF transmission, require attention due to their involvement in the transmission of Wuchereria bancrofti, which exhibits both nocturnal and diurnal periodicity. Consequently, we deemed it essential to assess mosquito sampling tools for all vectors to enhance LF elimination efforts. In 2022, the World Health Organization (WHO) estimated that there were 120 million people infected with W. bancrofti and 12 million people infected with Brugia malayi. This disease is most prevalent in tropical and subtropical regions, with approximately 80% of all cases occurring in Africa (World Health Organization, Global Programme to Eliminate Lymphatic Filariasis, Progress Report 2023, Geneva: World Health Organization; 2023). These statistics have been included in the revised manuscript (paragraph 1, page 4). Therefore, we had to omit studies related to Brugian filariasis, and as a result, we excluded relevant databases from our review.

Comment 2: While the manuscript provides an overview of various mosquito trapping techniques, a more detailed description of these methods would offer deeper insights and a better understanding of their utility. Although the study assesses trapping techniques for Aedes and Anopheles mosquito vectors, a more thorough description of these methods, along with data on the number of mosquitoes captured using each, could be presented in a more organized manner to enhance understanding and facilitate recommendations for improvement.

Response:  In accordance with the reviewer's recommendations, we have included descriptions and principles of various types of traps for mosquito collection in the revised manuscript. 

Reviewer 2 Report

Comments and Suggestions for Authors

The authors provide an informative and thorough comparative review of alternative trapping tools for Aedes spp. and Anopheles spp. vectors of lymphatic filariasis in various endemic regions. The manuscript is well written and conveys information that contributes to the knowledge base regarding targeted mosquito trapping methods/techniques. A few specific corrections/revisions to be made include:

1. Title - Italicize "Aedes" and "Surveillance" should not be italicized.

2. Figure 1 and Table 1 provide the same information. Suggest eliminating Figure 1 as it is too crowded with very small font size, thus difficult to read.

3. Table 2 - Realign column width to eliminate truncation, and correct punctuation (e.g., "et al.").

4. If listing genera provide species name or "sp." if singular, or "spp." if multiple species are referred to.

Author Response

We express our sincere gratitude to the reviewers for their invaluable suggestions. We have thoughtfully reviewed all of their recommendations and made the necessary revisions to the manuscript accordingly.

The authors provide an informative and thorough comparative review of alternative trapping tools for Aedes spp. and Anopheles spp. vectors of lymphatic filariasis in various endemic regions. The manuscript is well written and conveys information that contributes to the knowledge base regarding targeted mosquito trapping methods/techniques. A few specific corrections/revisions to be made include:

Comment 1. Title - Italicize "Aedes" and "Surveillance" should not be italicized.

Response:  We appreciate the reviewer for bringing this to our attention. As per the suggestion, we have implemented the necessary corrections in the revised manuscript.

Comment 2. Figure 1 and Table 1 provide the same information. Suggest eliminating Figure 1 as it is too crowded with very small font size, thus difficult to read.

Response: Yes, we concur with the suggestion, and accordingly, we have removed Figure 1 from the manuscript.

Comment 3. Table 2 - Realign column width to eliminate truncation, and correct punctuation (e.g., "et al.").

Response: As recommended, we have realigned the width and rectified the punctuation in the revised manuscript.

Comment 4. If listing genera provide species name or "sp." if singular, or "spp." if multiple species are referred to.

Response:  In accordance with the recommendations, we have implemented the suggested changes in the revised manuscript. 

Reviewer 3 Report

Comments and Suggestions for Authors

Article is not clear. It is a documentary research in which there is no order, firstly, there is no agreement between the abstract anf the content of the manuscript. The study considered a systematic review on prevalence, geograhical distribution and bio-ecology of filariasis, as a second point they will investigate and compare the effectiveness of capttures and define as primary objetive to define the best trap for capture. According the manuscript the main objective was the comparasion and effectiveness of traps for vector capture, but the sequence of the text does not show it that way

Furthemore, the abstract only refers to capture methods, nothing about prevalence, geographical distribution or bio-ecology.

In the description of the analysis they do it qualitative, values are required to see the impact of effectiviness, therefore several doubts arise that we explain below: Was it the same number of traps? To make the comparasion; Was the same periodand/or time of year when the sampling was carried out?, the traps were placed at same distances from posible food source.

It is very important to consider the study conditions and/or characteristics are as homologate as posible to give the same opportunity to the capture elements.

Furthermore, there was no statistical analysis to define difference among traps and test of means to make the sequence of the most effective al least.

The references section should be reviewed in depth in several of them the scientific name is misspelled, in many of them after writing the authors write words et al, which should not be in this part of the manuscipt.

Other comments:

-Title: According to what was written, the title does not coincide with the purpose and objectives mentioned in the writing, the systematic review.

-page 2 -
1.1. Prevalence and Distribution of Lymphatic Filariasis - paragraph 2: "compared to a high prevalence of 25.2% on Taveuni. Beye et al. (1953)" - If you are quoting with numbers, why write last names here?

-page 3 - "
The Anopheles type, in tropical Africa, Anopheles gambiae, Anopheles funestus" - Just as it abbreviates Culex, it should give the same shape to Anopheles

-page 3 - "
In south Pacific Islands, sub-periodic W. bancrofti is primarily transmitted by Ae. pseu- doscutellaris [39] Symes et al . [40] [22]" - the quote looks incomplete

-page 3 - "Burnett [41], Rossen [42] and Symes [22, 40]" -Because adding the surnames if already with the number it will be easy to locate it in the cited literature section

-page 3 - "
particularly in areas where members of the scutellaris and kochi groups have been reported [43]." - the kochi group has already been mentioned before, while scutellaris has not been, when mentioned for the first time the name must be complete     -page 4 - first paragraph - "Nelson  listed 26 Anopheles species"- please mention the year

Author Response

We express our sincere gratitude to the reviewers for their invaluable suggestions. We have thoughtfully reviewed all of their recommendations and made the necessary revisions to the manuscript accordingly.

Comment 1: Article is not clear. It is a documentary research in which there is no order, firstly, there is no agreement between the abstract and the content of the manuscript. The study considered a systematic review on prevalence, geographical distribution, and bio-ecology of filariasis, as a second point they will investigate and compare the effectiveness of captures and define as primary objective to define the best trap for capture. According, the manuscript the main objective was the comparison and effectiveness of traps for vector capture, but the sequence of the text does not show it that way

Response: We are confident that the esteemed referee will agree that mosquito interactions with their environment include their biology, behavior, and ecology. This information is essential for developing effective trapping methods. For example, different mosquito species have different biting habits, so traps must be designed to attract and capture the specific species of concern. We presented a discussion of distribution, physiological races, and bio-ecology to set the stage for reviewing sampling strategies in LF endemic areas (Aedes and Anopheles vectors). Therefore, we presented the results of the systematic review first, followed by the results of the comparison of the effectiveness of different traps. We felt that this was the most logical way to present the information.

Comment 2: Furthermore, the abstract only refers to capture methods, nothing about prevalence, geographical distribution or bio-ecology.

Response: We appreciate the reviewer's suggestion. We have now included a paragraph on the prevalence, geographical distribution of the disease and bioecology of the vectors in the revised manuscript.

Comment 3: In the description of the analysis they do it qualitative, values are required to see the impact of effectiveness, therefore several doubts arise that we explain below: Was it the same number of traps? To make the comparison; Was the same period and/or time of year when the sampling was carried out?, the traps were placed at same distances from possible food source.

Response: We appreciate the reviewer's relevant query. In the fourteen articles we reviewed, the number of traps deployed and the timing of trap collections varied. There was no mention of the proximity of food sources to the traps, except for their general placement indoors or outdoors.

Comment 4. It is very important to consider the study conditions and/or characteristics are as homologate as possible to give the same opportunity to the capture elements.

Response: We understand the importance of having homogenous local study conditions and characteristics to measure capture outcomes accurately. However, our primary objective was to review the available information and explore alternative trapping tools for Aedes and Anopheles vectors in LF endemic regions.

Comment 5: Furthermore, there was no statistical analysis to define difference among traps and test of means to make the sequence of the most effective at least.

Response: We value the reviewer's pertinent comment. In the fourteen articles we reviewed, the variation in the number of traps deployed and the timing of trap collections in different eco-epidemiological settings introduced heterogeneity. Given this factor, statistical analysis may not effectively capture the differences. We consulted a statistician regarding this matter.

Comment 6: The references section should be reviewed in depth in several of them the scientific name is misspelled, in many of them after writing the authors write words et al, which should not be in this part of the manuscipt.

Response. We extend our gratitude to the reviewer for identifying these unintentional errors. As recommended, we have rectified all of these issues in the revised manuscript.

Other comments:

  1. Title: According to what was written, the title does not coincide with the purpose and objectives mentioned in the writing, the systematic review.

Response: We appreciate the reviewer's feedback. As suggested, we have revised the title to "Mosquitoes, Lymphatic Filariasis, and Public Health: A Systematic Review of Anopheles and Aedes Surveillance Strategies," aligning it with the study's purpose and objectives.

  1. -page 2 - 1.1. Prevalence and Distribution of Lymphatic Filariasis - paragraph 2: "compared to a high prevalence of 25.2% on Taveuni. Beye et al. (1953)" - If you are quoting with numbers, why write last names here?

Response: We appreciate the reviewer's feedback, and we have implemented the required corrections in the revised manuscript.

  1. -page 3 - "The Anopheles type, in tropical Africa, Anopheles gambiae, Anopheles funestus" - Just as it abbreviates Culex, it should give the same shape to Anopheles

Response: We appreciate the reviewer's feedback, and we have implemented the required corrections in the revised manuscript.

4    -page 3 - "In south Pacific Islands, sub-periodic W. bancrofti is primarily transmitted by Ae.  pseu- doscutellaris [39] Symes et al . [40] [22]" - the quote looks incomplete

Response: We appreciate the reviewer's feedback, and we have implemented the required corrections in the revised manuscript.

  1. -page 3 - "Burnett [41], Rossen [42] and Symes [22, 40]" -Because adding the surnames if already with the number it will be easy to locate it in the cited literature section.

Response: We appreciate the reviewer's feedback.

  1. -page 3 - "particularly in areas where members of the scutellaris and kochi groups have been reported [43]." - the kochi group has already been mentioned before, while scutellaris has not been, when mentioned for the first time the name must be complete

Response:  We appreciate the reviewer's input, and as recommended, we have included the full name of the mosquito species to ensure clarity and accuracy.

  1. -page 4 - first paragraph - "Nelson listed 26 Anopheles species"- please mention the year

Response: We appreciate the reviewer's feedback, and we have implemented the required corrections in the revised manuscript.